# The Importance of Chitosan Coatings in Dentistry

**DOI:** 10.3390/md21120613

**Published:** 2023-11-26

**Authors:** Anna Paradowska-Stolarz, Marcin Mikulewicz, Joanna Laskowska, Bożena Karolewicz, Artur Owczarek

**Affiliations:** 1Division of Dentofacial Anomalies, Department of Orthodontics and Dentofacial Orthopedics, Wroclaw Medical University, Krakowska 26, 50-425 Wroclaw, Poland; marcin.mikulewicz@umw.edu.pl (M.M.); j.laskowska@umw.edu.pl (J.L.); 2Department of Drug Forms Technology, Wroclaw Medical University, Borowska 211A, 50-556 Wroclaw, Poland; bozena.karolewicz@umw.edu.pl

**Keywords:** chitosan, dentistry, preventive dentistry, coated materials, biocompatible

## Abstract

A Chitosan is a copolymer of N-acetyl-D-glucose amine and D-glucose amine that can be easily produced. It is a polymer that is widely utilized to create nanoparticles (NPs) with specific properties for applications in a wide range of human activities. Chitosan is a substance with excellent prospects due to its antibacterial, anti-inflammatory, antifungal, haemostatic, analgesic, mucoadhesive, and osseointegrative qualities, as well as its superior film-forming capacity. Chitosan nanoparticles (NPs) serve a variety of functions in the pharmaceutical and medical fields, including dentistry. According to recent research, chitosan and its derivatives can be embedded in materials for dental adhesives, barrier membranes, bone replacement, tissue regeneration, and antibacterial agents to improve the management of oral diseases. This narrative review aims to discuss the development of chitosan-containing materials for dental and implant engineering applications, as well as the challenges and future potential. For this purpose, the PubMed database (Medline) was utilised to search for publications published less than 10 years ago. The keywords used were “chitosan coating” and “dentistry”. After carefully selecting according to these keywords, 23 articles were studied. The review concluded that chitosan is a biocompatible and bioactive material with many benefits in surgery, restorative dentistry, endodontics, prosthetics, orthodontics, and disinfection. Furthermore, despite the fact that it is a highly significant and promising coating, there is still a demand for various types of coatings. Chitosan is a semi-synthetic polysaccharide that has many medical applications because of its antimicrobial properties. This article aims to review the role of chitosan in dental implantology.

## 1. Introduction

Polymer coatings in dentistry are widely used and, recently, materials obtained from natural polymers have been widely treated as matrices for drug delivery, especially in surgery and periodontics [1,2,3]. The coatings may also be used in other dental specialities. In stainless steel corrosion, the elution of metallic ions from the surface of materials used in orthodontics, and which can be protected by coating, has been proven by Mikulewicz et al. [4]. This phenomenon has also been shown to influence gene expression [5]. In dentistry, coatings are used to improve the quality and properties of dental devices [6].

### 1.1. Chitosan Structure

Chitosan, a naturally derived marine polymer with linear amino-polysaccharide structure, is widely used in dentistry. A D-glucosamine and N-acetyl glucosamine copolymer is obtained by deacetylation of the chitinous exoskeletons of crustaceans (also known as shellfish scaffolds). It also occurs naturally in mammalian cells. The main characteristics of chitosan are biocompatibility, non-toxicity, regenerative properties, natural availability, and the possibility of chemical treatment. According to the Food and Drug Administration agency (FDA), it has biological activity properties and wide spectrum of usage against all types of bacteria, including Gram-positive and Gram-negative. Due to those conditions, it is used willingly to treat damages in the tissues of the oral cavity. Additionally, chitosan has elastane-like properties [1,7]. Chitosan represents a group of multifunctional excipients [8]. It is used as an auxiliary substance and an antimicrobial agent. This is especially important in current times due to the possible elimination of the SARS-CoV2 virus from saliva [9,10,11]. It is used in preventive and conservative dentistry, endodontics, periodontal procedures, surgery, prosthodontics and orthodontics [7,12,13]. Although widely used, it is still unclear whether chitosan could potentially be an allergen for humans [7]. Although chitosan is more frequently a drug carrier component and active formulation agent, the variety of its use changes [14]. It shows a high percentage of antimicrobial reactions, especially against *Streptococcus gordonii*, which is the first bacteria to colonise surfaces in the oral cavity [15].

In a linearly built chitosan molecule, the relationship between the present hydroxyl and amino groups forms hydrogen bounds that influence the structure and flexibility of the polymer chain. Chitosan is a copolymer of glucosamine and N-acetyl-D-glucose amine units as shown in Figure 1.

Natural polymers and natural-based medicine are prominent topics of recent research. It is important to search for accessible materials which present biocompatibility and biocompetence. Chitosan is one of the natural materials which meets these criteria. The authors of the presented study are engaged in the two different fields of medicine–pharmacy (BK and AO) and dentistry (APS, MM and JL). We therefore tried to combine two points of view on chitosan: coatings–pharmaceutic and dental. We found this approach to be the advantageous to our research.

### 1.2. General Use

Chitosan belongs to the group of multifunctional polymers used in dentistry. It is used to design carriers and materials adapted to some specific conditions, as well as the specificity of the application site, which deliver active substances directly to diseased changed tissues. This includes intrapocket application and the release of the active substances in a timely manner. This specification allows for the improvement of the form of application and further development of therapeutic strategies. Its multifunctionality results from the function of the auxiliary substance that allows for the development of both the mucoadhesive carrier and the formulation component with biological activity. This phenomenon was confirmed in the biological activity tests, the mentioned antibacterial, haemostatic, but also antifungal and immunomodulating properties of the polymer and its derivatives, i.e., chitooligosaccharides. In the literature, chitosan, in addition to its antibacterial, antifungal, and anti-inflammatory properties mentioned before, is also reported to act as a biocompatible dental carrier that accelerates the process of periodontal regeneration. The biological activity of the polymer is caused by the cationic nature of the polymer and the interaction of the positively charged amino-group molecules with the negatively charged components of the Gram-negative bacteria cell membranes. As a result of the interaction, the properties of the cell membranes of the bacteria change-the transport of the nutrients inside the cell fluctuates, and its content starts leaking outside. Another explanation for the polymer activity is its penetration into the bacterial cell interior and binding to the DNA of the bacteria, which inhibits RNA transcription and, consequently, handicaps the protein synthesis. The biological activity of the polymer depends on its physicochemical properties, e.g., molecular weight, degree of polymerisation, degree of deacetylation, and also depends on the type of the microorganism. The properties of the polymer are crucial for its activity. It has been proven to react strongly against the bacteria responsible for creating dental plaque, including *Porphyronomas gingivalis, Prevotella intermedia,* and *Aggregatibacter actinomycetemcomitans* (*formerly Actinobacillus actinomycetemcomitans*). Due to all of these characteristics, polymers find many biological applications in materials and carriers used in the treatment of periodontitis, bone and tissue regeneration, drug delivery, and wound healing. However, the choice of appropriate molecular weight and degree of deacetylation affects the desired application properties as well as development of the carriers, e.g., matrices, bioadhesive tablets, films, microspheres, microparticles, nanoparticles, nanofibers, or gel forms.

### 1.3. Molecular Interaction

The use of chitosan in dentistry is also due to its increased induction of osteoblast growth and binding around polymer coated implants, which allows for the obtaining of bioactive surfaces ensuring osseointegration. Research has shown that chemical modifications of the polymer structure increase its antibacterial activity and osteoinductive properties. The polymer has the ability to bond to the metals surface, which can improve their mechanical strength and increase the durability of, e.g., titanium implants. Some researchers, however, emphasise that the adhesion of chitosan coatings to titanium surfaces is not suitable for clinical applications. Ferraresse et al. [16] proposed a chemical treatment of the titanium implant surface, using the method of ‘direct coating’, which is simply immersing the surface in an acid solution that allows a strong electrostatic attraction between the chitosan particles and the coated surface. In this condition, stable coatings are obtained mechanically and chemically, and chitosan-coated dental implants can have increased possibilities for osseointegration, which can lead to their commercial use. The preparation of the implant surface and bone preparation affects the stability and clinical success of the dental implant. To improve osseointegration, a number of methods of surface treatment and implant coatings have been tried, with promising results in coating them with chitosan. Chitosan coating can affect the surface that comes into contact with bone by changing the biological, mechanical and morphological properties of the surface. For example, considering the mechanical properties, the chitosan coating changes the modulus of elasticity, thus reducing the mismatch of the implant surface with the bone of the alveolar process and reducing stress concentration areas [17,18]. In addition, chitosan coatings can potentially be used to release drugs, such as antibiotics, for local delivery around the implant area. Additionally, chitosan can prevent damage to the tooth surface by organic acids by buffering saliva pH in the mouth, preventing adhesion pathogens, and stimulating the structured regeneration of oral soft tissues.

### 1.4. Biomechanical Features

Chitosan and chitosan-based coatings have relatively low stiffness and strength, especially in high humidity. Therefore, it is valid to obtain modified chitosan sol-gel preparations to improve their mechanical properties. The addition of chitosan improved the antibacterial activity of two types of subjects [9,19] of hybrid silica-chitosan coatings (50M50G and 45 M45G10T) and increased the proliferation rate of cells that grow on their surface. The proliferation indicators were the highest for coatings containing 5% and 10% chitosan. We can also conclude that the addition of chitosan and TEOS modulates the release of Si, which plays an important role in osteocompatibility. Tertiary coatings containing 5% and 10% chitosan have very good antibacterial properties [20]. They should be suitable for dental implants, as they protect against bone infection for a long time. The introduction of these materials into dental practice can increase the number of patients admitted to implantation. 

A study by Tarsi et al. [21] showed that chitosan is an interesting candidate capable of preventing the adhesion of bacteria *Streptococcus mutans* to hydroxyapatite in teeth with tooth decay. This effect was attributed to the ability of chitosan to stimulate the orderly regeneration of soft tissues of the oral cavity, preventing the harmful effects of organic acids and the bactericidal effect. The desorption effect of chitosan was weaker when *Streptococcus mutans* adhered to saliva-coated hydroxyapatite in the presence of sucrose. These results indicate the possibility that the presence of small amounts of chitosan in the teeth, mouthwashes, or chewing gum may interfere with the bacterial colonisation of the surface of the teeth. The use of chitosan in dentistry is widespread and reflects many branches of this discipline. Table 1 presents examples of methodology and results of chitosan coatings used in dentistry. In summary, chitosan is a semisynthetic polysaccharide that has many medical applications due to its antimicrobial properties.

### 1.5. Aims

There are existing reviews on chitosan and its use in dentistry in general, while the literature is expected to focus on a more particular application in dental specialities. Existing reviews attempt to answer the question: What are the general results of dental treatments using chitosan and does the use of chitosan provide beneficial clinical results? [30] Furthermore, Hallmann et al. [31] evaluated chitosan’s properties for use in dental implantology, and Qu et al. [32] investigated the use of chitosan as a biomaterial for the prevention and treatment of dental caries. Researchers believe that developing chitosan-based solutions in dental implantology, orthodontics, and prosthetics will improve clinical practice. The purpose of this review was to identify gaps in existing studies for potential future research.

## 2. Search Criteria

### Methods

Pubmed (Medline) database was searched by key words “chitosan coatings” and “dentistry” in August 2022. The search was carried out according to the PRISMA guidelines. In total, 135 articles were found, among them 131 full text articles were selected. To gain the most recent data, the filter excluding papers published earlier than 10 years ago and English-written papers was used, which gave us a number of 112 papers in total. After including the words “Covid-19” or “SARS-CoV-2”, no results were found for this investigation. We checked this because the pandemic was one of the most important topics of scientific research in the last 3 years. After skimming through the abstract, the papers that did not fit the topic were excluded, which gave us the final number of 92 papers. One of the articles was duplicated. Furthermore, the obtained full text articles were read, and additional articles were excluded, especially those concerning influence of chitosan coatings on the genome and compositions added to the dental materials. In the end, 23 papers were chosen that concerned the topic directly.

## 3. Biomedical Application of Chitosan Coatings

Chitosan is recognized as an exciting and promising excipient for the pharmaceutical industry. There is evidence that the addition of drugs, chitosan nanoparticles can improve absorption in the whole small intestine, increase survival time, reduce drug cardiotoxicity, reduce tumour cell dispersion, and even improve tumour cell destruction efficiency. Chitosan nanoparticles prevent proteins and DNA from being degraded by gastric enzymes and nucleases, thereby increasing the resident time in the gastrointestinal tract. This leads to greater control of drug release, improved protein biodegradation, and increased absorption of hydrophilic molecules by the epithelium layer. These possibilities have also been used for insulin, resulting in high drug entrapment efficiency, good stability, low outbreak, and consistent insulin release [33].

Moreover, chitosan-based nanoparticles are used to improve vaccine delivery due to their mucoadhesive and osmotic properties, which increase peptide adsorption and transport through the nasal epithelium. Chitosan nanoparticle glucuronidation significantly increased systemic (serum immunoglobulin G (IgG) titers), mucosal (secretory immunoglobulin A (IgA)), and cell-mediated (IL-2 and interferon- (IFN-)) immune responses. In addition, there is evidence that chitosan can stimulate an immune response on its own [34].

Furthermore, chitosan and its derivatives have been found to have haemostatic potential, antimicrobial activity, and biocompatibility. In addition, chitosan is effective in lowering plasma and liver lipid levels in rats fed high-fat diets and increasing lipase activities [35]. Furthermore, it prevented and improved the condition of hypercholesterolemia in rats fed a high-fat diet. Longer term feeding of chitosan resulted in a better hypolipidemic effect [36]. Chitosan induced the formation of granulation tissue or re-epithelialization in the early stages of wound healing, so it is commonly used as a wound care material [37].

## 4. Dental Application of Chitosan Coatings

### 4.1. Surgical Procedures

Chitosan coatings, because of their properties in increasing tissue healing and stabilisation of implants and membranes, are mainly used in surgery, and they could be used in other dental specialities. This is thanks to its potential to increase healing and antiseptic activity. Chitosan coatings are used especially during dental implantation. The search for a perfect coating agent that has a bacteriostatic effect, which also results in greater implant stabilisation, was implemented. The most common problem with implant loss is periimplantitis around the metal part of the implant and the need for better osteointegration pairs with the need for an antimicrobial reaction. It has been proven that chitosan/AgNPs and chitosan/HA coatings protect the surface more effectively, especially when used on porous surfaces [38,39,40]. VEGF (vascular endothelial growth factor) plays a role in vascularization and osteoblastic differentiation as well as bone regeneration. These procedures are key factors in the osteointegration of dental implants. There are serious trials to add a chitosan coating to the implant surface to increase the osteointegrative process, although the results of this research have not been published yet [41]. The addition of other substances, however, increases osteointegration with implants [42]. The addition of chitosan is even more effective when the implant surface is under osteoblast formation [43]. Recent interest in the application of chitosan in dental implantology is also the key to the attention of other researchers, which is presented in the recent review by Hallmann et al. [31]. One of the most specific branches where chitosan coatings are used is dental and orofacial surgery. The substances and the use of chitosan-based coatings in surgery are presented in Appendix A, [15,16,44,45,46,47,48,49,50,51,52,53,54,55,56,57,58] which is in the supplementary files.

### 4.2. Disinfection

Chitosan finds a wide range of uses as a disinfection agent, especially when prosthetic restorations and devices should be disinfected to increase wound healing [50]. Recent studies show that alternative methods of disinfection are being researched [59]. Thanks to the possibility of creating a coating, chitosan might be beneficial in the use of that agent for a longer period of time when compared to conventional disinfection agents. Additionally, the addition of chitosan to hand sanitisers increases the bactericidal effect [60].

### 4.3. Dental Appliances

Among dental appliances, we could find removable prostheses, orthodontic appliances, and occlusal splints [1]. The influence of the archwire coating on the properties of the archwire and what follows: friction, duration of treatment, and risk of complications is widely known [61]. When it comes to chitosan, the benefits of using coatings are quite large. Chitosan coating reduces friction and therefore helps achieve better final bracket expression and reduce treatment time [62]. It could also be beneficial due to the bactericidal activity, because the orthodontically treated patient may have some problems achieving perfect oral hygiene.

Chitosan coatings could be used in postoperative prosthodontics, where the denture is applied to the patient’s mouth directly after the surgical procedure [63].

It had been proven that chitosan coating of the dental prostheses has its antifungal properties, especially against the most commonly observed in the oral cavity *Candida* spp. [64]. It may be extremely important not only on the surface of the appliance itself, but also when the impression is transferred between the dental office and the technician [59].

Except for antifungal, haemostatic and antibacterial properties, chitosan could be used to cover the tooth surfaces, e.g., as an ingredient of the toothpaste. As a coating on the surface of the tooth, it desensitises the tooth to external conditions, such as acids in food [61]. 

Although chitosan finds less obvious outside of surgical and periodontological procedures, the application of its coatings has been collected in Table 2. 

### 4.4. Risk of Bias

It was not possible to assess any univocal statistics among the examined papers, as the papers apply to different methods and applications. For this reason, the article was prepared as a narrative review. 

## 5. Advantages and Limitations of Chitosan Coatings in Dentistry

Due to the many possibilities of the use of chitosan in dental applications, its use in general is beneficial. By binding those two substances, we could gain perfect novel drug carriers or create new drug formulations. Biodegradability, biocompatibility, antimicrobial activity, and low immunogenicity are properties that are advantageous in terms of the prevention and treatment of inflammatory diseases and can accelerate the development of biological materials for wound healing and osteointegration in dentistry. The mixture of natural polymers with nanomaterials could be a future design of dental materials. The fact that the material is naturally-based makes it interesting to the researchers and keeps on developing the new trends to its use, not only separately, but also as a part of materials and matrices used in clinic. One of the examples of that use was the addition of binding nanomaterials with chitosan in endodontic treatment. By binding those two substances, we could gain perfect novel drug carriers or create new drug formulations. Although the use of chitosan itself showed a beneficial effect in endodontic treatment, its application with nanomaterials also seems to have great possibilities [12,66]. This proves that the cooperation of material engineers, pharmacists and dentists is crucial to introducing new potential applications of chitosan. 

The problem of the biocompatibility of dental materials used in dentistry is very large, especially due to the fact that this could prolong the retention of teeth in the dental socket. According to mimicking, this is the most natural way of healing with such materials [67,68]. Also, the trend to use more natural substances and the development of green dentistry make naturally-found substances like chitosan important [69]. The development of 3D printing in dentistry also adds new perspectives in the world today, as the materials are biocompatible and more precise in preparation [70]. In addition, the porous structure of the material creates a similar structure to natural matrices that mimics the adhesive properties of the cells. This structural construction emphasizes the three-dimensional distribution of the substance particles, creating a healing process and allowing for bone regeneration and new bone formation. The three-dimensional distribution refers not only to dentistry but also to all aspects of medicine and science [71,72,73,74]. Chitosan has many applications in almost every field because of its unique properties, but it also has some limitations. It is insoluble in water and most organic solvents, limiting its applications in a variety of fields. It is insoluble in water as a result of the presence of intermolecular and intramolecular H-bonding. It can only be dissolved in dilute acids like 1% acetic acid, formic acid, lactic acid, and so on. However, several modifications can be made to improve solubility [75].

## 6. Future Directions

Given that the surfaces of dental materials used after coating may change, one has to predict how to make the use of the dental materials as repeatable as possible. The nanostructure and microstructure of the materials changes. Even a 1 m layer of 1 µm of the coating can have an effect on properties and act in an antibacterial manner, which is a promising for dental implantology [76,77,78]. Other natural polymers could also be searched to be used as coatings, which could also be promising in the treatment of serious conditions, such as oral cancer [1,79]. Because of the high biocompatibility, the scaffolds could heal into the tissues and give great potential in tissue restorations. The experiments performed on rabbits show that Mg-CD/CH coated Ti-6Al-4V helps in bone regeneration procedures and reduces the bone defects. This could open the door to future bone scaffolds in tissue engineering and gives prospects for the production of novel bone substitutes [80]. Although chitosan is a very important and promising coating, different types of coatings to help heal and osteointegrate are still needed and being researched. Recent studies have hydroxyapatites (HAPs) are even more effective in fighting implant loss and that the bacteriostatic reaction is greater compared to chitosan [81]. Chitosan coatings could be added to help heal alveolar bone loss and during implantation to increase the possibility of implant adaptation and healing. In the authors’ opinion, the studies with chitosan coatings should go into the fractal dimension analysis of bone to visualise the actual bone condition around the dental implant [81].

## 7. Conclusions

The study presented shows that chitosan has many applications in dentistry. Due to its antimicrobial effect and natural origin, it is a perfect natural disinfection agent. It is used not only in dental surgery, including implantology, but also in other dental specialities, including conservative dentistry, orthodontics, prosthetics etc. Chitosan is a multifactorial naturally-derived polymer that is very useful in pharmaceutical technology. Due to the multidisciplinary approach of this paper, we have shown that the cooperation between pharmacists and dentists might be useful in fully understanding the development of its use. Its natural origin makes chitosan a perfect candidate for multiple applications in the environmentally-conscious thinking of the world today.

## Figures and Tables

**Figure 1 marinedrugs-21-00613-f001:**
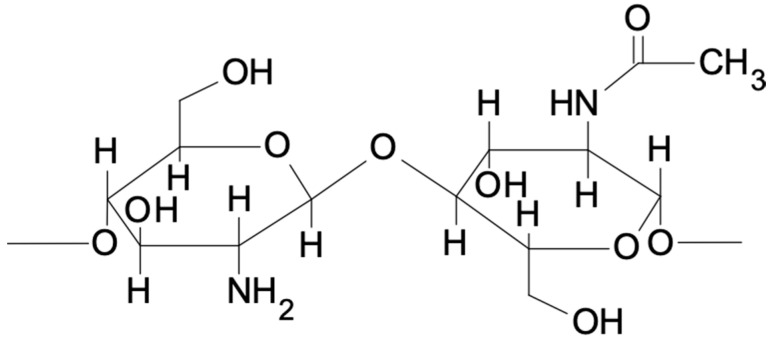
Chemical structure of chitosan.

**Table 1 marinedrugs-21-00613-t001:** Obtaining chitosan coating: methods and basic uses.

Coating	Material	Results	References
Chitosan coatings (80.7% deacetylated polymer, 108 kDa) with incorporated tetracycline (20%) or digluconate chlorhexidine (0.02%) applied to the titanium implants in reactions silane	- Implants were covered with 3-aminopropyltriethoxysilane at pH 4.3, dried, then 2% glutaraldehyde was added for obtaining a reactive group aldehyde, which reacts with amino groups in chitosan particles - Tetracycline or chlorhexidine was mixed with chitosan in 1% solution of acetic or formic acid - The chitosan solutions were poured out on titanium surfaces coated with glutaric silane-aldehyde and dried in room temperature for ~3 days, rinsed by a 0.05M NaOH wash to neutralize the residual acid solvent - The samples were sterilized with ethylene oxide at 38° C for 5 h	- The coatings released 89% of the tetracycline by 7 days and 100% of the chlorhexidine by 2 days - The released tetracycline inhibited 95%-99.9% of pathogen growth up to 7 days without cytotoxicity to human cells - The released chlorhexidine was active against pathogens for 1 to 2 days (56%–99.5% inhibition), and its cell toxicity was observed on the first day after the application. The potential for local delivering antimicrobial substances was observed; its purpose was to inhibit bacterial growth without toxic effects to host cells/tissues - Release of tetracycline and chlorhexidine from the coatings were sufficient to stop bacterial growth for a few days, which can provide the necessary time for the normal host cells to stick to the implant surface and inhibit bacterial growth - Chitosan coatings may be used as a local drug delivery system for antibiotics	Norowski et al. [22]
Chitosan coating (92.3% deacetylated polymer)	- 1% chitosan solution in 1% acetic acid was poured on ground titanium pins (rough ground titanium pins) covered with 3-aminopropyltriethoxysilane - As controls uncoated titanium pins and coated with calcium phosphate were used - In the tibias of 16 adult white male rabbits from New Zealand, 2 chitosan coated pins and 1 covered with phosphate calcium and uncoated were implanted - After 2, 4, 8 and 12 weeks histologically an angle healing and bone formation were assessed	- Histological assessments of tissues in contact with chitosan coated spikes showed minimal inflammatory and typical response sequence of fibrous healing, bone formation followed by the development of the lamellar bone patterns of healing and development of bone connection with implant were assessed around chitosan-coated implants in a 12-week bone model, the tibia of the rabbit was similar to those in coated titanium pins and uncoated with calcium phosphate - Chitosan coatings can offer benefits in drug and growth factor delivery, compared with traditional ones, coated with calcium phosphate and uncoated implants - Chitosan coatings provide osseointegration of dental implants	Bumgardner et al. [23]
chitosan coating	- titanium implants coated with chitosan through dipping in the prepared polymer solution, then dried in a dryer at 25 °C in order to produce a uniform 50% relative humidity films to avoid cracking and shell deformation - sterilized with gamma radiation	- In the jaws of dogs, four implants were put and 12 weeks after the surgery euthanasia was performed - excised bone blocks were taken and assessed for with the help of computed microtomography and two parameters of the bone were measured: bone contact with the implant surface (BCIS) and bone area around the implant (PIBA) - the results confirmed the suitability of the chitosan coatings on titanium surfaces in improving the osseointegration of dental implants	López-Valverde et al. [24]
Chitosan coated liposomes	- Liposomes prepared forLipid base: Egg-phosphatidylcholine, Nitrobenzoxadiazol-4-yl-phosphocholine, fluorescent phospholipid labelled with fatty acid and lipids, egg L-α-phosphatidylglycerol or Dioleyl-trimethylammoniumpropane forming a lipid film by rotary evaporation of lipid solution in chloroform, then a phosphate film buffer was added (pH 6.8) - Size reduction was obtained by extrusion of double-layer polycarbonate membranes at size 200 nm - The coating was obtained through electrostatic deposition after mixing the liposomal dispersion with the solution polymer	- Positively charged liposomes showed higher adhesion ability to hydroxyapatite compared with liposomes negatively charged, but positively charged liposomes aggregated in artificial saliva - Can be used in patients with reduced salivation	Pistone [25]
Titanium alloy implant (Ti6Al4V) coated chitosan	- Implants were covered with 3-iso-cyanopropyltriethoxy silane at pH 4.5–5.5, then 2% glutaraldehyde was added to give a reactive group aldehyde, which reacts with amino groups in the chitosan particles - The implants were immersed in chitosan solution in acetic acid at 4 °C, then the excess water evaporated for more than 7 days - The implants were rinsed with NaOH solution, then water	- Cell adhesion and proliferation of the fibroblasts for the implants have been improved, and simultaneously bacterial proliferation has been inhibited by the chitosan coating - It has been shown that chitosan can be used as material for coating areas in periodontal healing of dental implants	Kalyoncuoglu et al. [26]
Chitosan and polycaprolactone (PCL) multilayers coating for metallic implants with incorporated vancomycin or daptomycin in microspheres with poly (methyl methacrylate) (PMMA) for treatment of periimplantitis	- cCoating produced with dip-coating technique	- Studies have shown that drug release in case for both types of drugs occurred by diffusion, and the release profile depended on the type of the drug, the pH of the solution, and whether the drug was incorporated directly into the film coating, or encapsulated in PMMA microspheres - Coatings containing daptomycin directly incorporated into the films released 90% of substances after 1 day at pH 7.4 and after 4 days at pH 5.5 - Films with microspheres with incorporated daptomycin achieved 90% release after 2 days of treatment time at pH 5.5 and after 2 days at pH 7.4 - Coatings containing vancomycin directly incorporated into the films showed 90% of drug release after 20 h at pH 5.5 and 2 and 3 days respectively at pH 7.4 - Films containing daptomycin showed antibacterial activity against both MRSA and susceptible strains of *S. aureus*, which confirms the possibility of use of such films as coatings for the relief of *S. aureus* infections around the metal implant	Soares et al. [27]
Polylactic-co-glycolic acid (PLGA) nanoparticles coated with chitosan hydroxypropyltrimonium chloride as a carrier for rebamipid	- It was received positively charged nanoparticles with mean particle diameter of 97.0 ± 36.7 nm	- It was confirmed that the mucin adsorption capacity by nanoparticles coated with chitosan was 2.3 times higher than in uncoated nanoparticles - Therapeutic efficacy of nanoparticles was evaluated in the treatment of a mouse model stomatitis, induced by cancer chemotherapy - In the group administered with nanoparticles covered with chitosan, the ulcer area decreased significantly on the 9th, 11th and 13th day when compared to an untreated control group; additionally, the treatment period was significantly shortened by 3.6 days when compared to the group in which uncoated nanoparticles were administered - Chitosan coated PLGA nanoparticles containing rebamipid may be beneficial in treatment of oral mucositis, resulting from cancer chemotherapy	Takeuchi et al. [28]
Gold nanoparticles (AuNP) coated chitosan-grafted thymol (CST)	- Implantation of thymol into the chitosan skeleton synthesized via adaptation of the Mannich reaction - AuNP particles were obtained mostly in spherical shape and medium-sized 2.41–3.30 nm	- CST coating on the surface of nanoparticles has been successfully used against cariogenic bacteria in the mouth - Electrostatic properties of CST used primarily for stabilization of the AuNP - CST coating on the AuNP surface has potential to be used in fighting infections caused by cariogenic bacteria	Pakawat et al. [29]

**Table 2 marinedrugs-21-00613-t002:** The use of chitosan coatings in other than surgery and periodontology specialities.

	Use	Author (Reference)
Prosthodontics	Antifungal properties	Jung et al. [64]
Conservative dentistry	Desensitizer, used to coved tooth surfaces in tooth pastes	Cicciù [30]
Orthodontics	Helping with healing after miniimplants placement	Anggani [42]
	Lower friction and therefore faster tooth movement in fixed appliances; better root control movement; reduced treatment time; better anchorage; reduced risk of root resorption	Elhelbawy [62]
Endodontics	Improved reaction anti-bacterial species *Peptostreptococcus* and *Fusobacterium*	Asadi [65]

## Data Availability

No new data were created or analyzed in this study. Data sharing is not applicable to this article.

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
