# Peer review of "The Importance of Chitosan Coatings in Dentistry"

_marinedrugs, 2023, doi:10.3390/md21120613_

Round 1

Reviewer 1 Report

Comments and Suggestions for Authors

Dear authors,

I congratulate you for the meticulous work done in your article "The importance of chitosan coatings in dentistry".

In my opinion, your narrative review is able to illustrate the characteristics of Chitosan and its clinical applications in a complete and easy to understand way.

The literature review, likewise, is sufficiently extensive and properly conducted, as is the literature referenced in the article.

I therefore believe that your work certainly deserves to be published in "Marine Drugs" in its present form.

Best regards

Author Response

Dear Reviewer 1,

thank you for the positive feedback on our research. Due to the other two reviews, we decided to check the language of the paper. Thank you. 

Reviewer 2 Report

Comments and Suggestions for Authors

A review paper describing the use of chitosan in dentistry, if well organized and clearly discussed, can be of interest in the biomedical and clinical fields.

Here, the Authors say (ABSTRACT).“This narrative review aims to discuss the development of chitosan-containing materials for dental and implant engineering applications, as well as the challenges and future potential.”

However, the article, in the present form, does not meet the aims since it is confusedly organized and not clearly discussed. In addition, it contains repetitions and the quality of the English language needs deep revision.

MAJOR REMARK: Being a review article, the level and completeness of acquisition of relevant literature is of primary importance. Was the development of this study based on the PRISMA guidelines (http://www.prisma-statement.org/)? Besides PubMed, the following databases are usually used: Web of Science, Scopus, Science Direct, EBSCO.

MAJOR REMARK: Being a review article, the division into sections and subsections must be consistent and appropriate to the subject matter. Here, an Introduction is followed by a “Results” section, which is quite unusual for a review article. In addition, the Introduction appears to be dedicated to the description of chitosan (structure, properties and uses in general). Why include Table 1 in the Introduction? What is the difference between Table 1 and Table 2? Figure 1 is wrong (the two co-monomers have the same chemical formula and the caption of this figure has to be more descriptive), whereas the sentence preceding Figure 1 contains errors and does not explain the possible relationships between the two different monomers in chitosan.

About point 1.3 (Molecular interaction) can you explain why “considering the mechanical properties, the chitosan coating CHANGES THE MODULUS OF ELASTICITY, thus reducing mismatch of the implant surface to the bone…”? Can you provide bibliographic references?

At point 1.4 (Biomechanical features), please explain why “…it is valid to obtain modified chitosan sol-gel preparations to improve its mechanical properties”? For what follows (from “the addition of chitosan” until “have very good antibacterial properties”), it is important to provide precise bibliographic references. Also, what is the meaning for “Coatings TERNARY”?

In addition to the observation on the meaning of both Tables 1 and 2, if Table 1 should persist, it  can profitably be made more concise.

About section 2 (Results), it should be titled differently and differently described, so to become more relevant to the purposes of the review.
“Materials and Methods” is only “Methods” and will describe how the data were collected from literature, selected and listed

The following sections should describe the different applications of chitosan in dentistry, divided by type, in a logical and easily readable way, without repetitions

The Title of Table 2 is hardly understandable, considering the data presented, are we talking about only coatings? Are the applications different from those of Table 1?

Note that the numbering starting from Reference [32] is wrong (Halmann et al. has the same number of Karimi et al.)

Comments on the Quality of English Language

The English language presents many spelling mistakes and some poorly constructed sentences

Author Response

Dear Reviewer 2, 

thank you for an effort to correct the paper. In the attachment we add more detailed replies to your comments. We hope that the article now corresponds to the suggestions of yours. Thank you once again.

Reviewer 3 Report

Comments and Suggestions for Authors

This is a narrative review on the use of chitosan coatings in dentistry. I accepted the review invitation because this topic is relevant and I am interested in this issue. Nevertheless, I apologize but the efforts of the authors to do this review were not enough to make it either captivating for the reader or relevant for the field of research. I hope authors can dedicate more time and effort to increase the level of this manuscript to a higher level. At this stage, it is hard to read or find interest in this review.

The main issues with this manuscript are the fact that we cannot identify the gap in knowledge that the study is intended to address and/or the limitations of previous reviews in the area. The organization of the introduction is also questionable; using subheadings for the introduction does not help to create an exciting flow of ideas; the use of a table to exhibit previous coating methods shows low level of time dedicated to the synthesis of the existing information; and finally, the aim presented is too ambiguous. In other words, at introduction, authors need to present knowledge to the readers, not raw data.

At results section is not comprehensible what is the connection between the topic of this review and the “Covid-19”. Why is Covid-19 introduced at the search strategy if it was not mentioned at all at introduction section? 

Reference numbers in the table 2 are not in agreement with the numbers in the reference list (Paulino-Gonzales et al. [34]; Caldeirao et al [35]. Arias et al [36]) and so on...

There is no rationale in the presentation of the results, lots of concepts are mixed up and spelling errors denote lack of care in preparation of the manuscript. Native English correction is advisable.

Risk of bias is not supposed to be applied in a narrative review like this one, only in systematic reviews, which is not the methodology of this manuscript 

Discussion should be used to interpret and explain the results. Both the strengths and weaknesses of the observations should be discussed. 

Comments on the Quality of English Language

Native English correction is advisable.

Author Response

Dear Reviewer,

thank you for an effort. 

Please, find our comments and response in the uploaded file. 

Best regards.

Round 2

Reviewer 2 Report

Comments and Suggestions for Authors

I thank the Authors for their effort in revising this review article in accordance with my observations and suggestions

I have no other comments to add

Comments on the Quality of English Language

Although some corrections have been made, the English still needs editing

Author Response

Dear Reviewer,

Thank you for the comment and the time you spent on the review of our manuscript. The English had been reedited - if that was not enough, we will do the second round of corrections. 

Best regards - Authors

Reviewer 3 Report

Comments and Suggestions for Authors

The concerns raised in the first review were not addressed by the authors.

Comments on the Quality of English Language

English needs to be reviewed by native speakers.

Author Response

Dear Reviewer,

thank you once again for the review of our manuscript. We tried to follow your suggestions for the second time. We hope this would satisfy you more. If not, please give us more detailed information on how to improve the paper. Thank you for understanding in that point. Best regards - Authors.

Round 3

Reviewer 3 Report

Comments and Suggestions for Authors

Authors performed the asked corrections.